# Simulation for Cu Atom Diffusion Leading to Fluctuations in Solder Properties and $Cu_6Sn_5$ Growth during Multiple Reflows

**Min Shang [1], Chong Dong [1], Haoran Ma [2,\*], Yunpeng Wang [1] and Haitao Ma [1,\*]**

1  School of Materials Science and Engineering, Dalian University of Technology, Dalian 116024, China; shangxiaomin2019@mail.dlut.edu.cn (M.S.); dongchongwyy@gmail.com (C.D.); yunpengw@dlut.edu.cn (Y.W.)
2  School of Microelectronics, Dalian University of Technology, Dalian 116024, China
\*  Correspondence: mhr@dlut.edu.cn (H.M.); htma@dlut.edu.cn (H.M.)

**Abstract:** The multiple reflows process is widely used in 3D packaging in the field of electronic packaging. The growth behavior of interfacial intermetallic compound (IMC) is more important to the reliability of solder joints. In this paper, experimental measurement combined with simulation calculation were preformed to investigate the evolution of Cu concentration in solders during multiple reflows, as well as its effects on the growth behavior of IMC and solder properties. The concentration of Cu in solder fluctuated, increasing with the increase of reflow times, which led to the fluctuation in the growth rate of the IMC. Furthermore, the Vickers hardness and melting point of the solder fluctuated during the multiple reflow processes due to the fluctuation in the Cu concentration. The data generated during this study could help to develop machine learning tools in relation to the study of interfacial microstructure evolution during multiple reflows.

**Keywords:** multiple reflow; Cu concentration; $Cu_6Sn_5$ growth; solder property

## 1. Introduction

Through the development of technology, the assembly density of packaging bodies is increasing, and the packaging process is becoming increasingly complex. It is difficult to complete the entire structure of the encapsulation process through a single reflow [1,2]. Three-dimensional electronic packages, characterized by the miniaturization of electronic products, generally require multiple reflows to complete their interconnection needs [3–5]. In particular, multiple reflow is an inevitable and commonly used packaging process in 3D electronic packaging technology, spawned by the trend towards the miniaturization of electronic products [6,7]. One example of this is the soldering processes of the "Double-POSSUMTM" 3D packaging structure developed by Amkor, the A9 application processor fabricated by Apple or the AMD's graph card made by Hynix [8–10]. In this package, the three daughter chips are first interconnected with the larger mother chip in the form of a flip-chip to form the device, before being interconnected with the largest mother chip, then connected to the package substrate and, finally, interconnected with the PCB substrate to complete the package. In this 3D packaging structure, multiple flip-chip technologies are used, which inevitably leads to multiple reflow processes. Therefore, the maturity of the multiple reflow process has a significant impact on 3D packaging technology. The core scientific issue of the reflow process is the growth mechanism and control factors of the IMC layer at the liquid/solid interface, which is of great theoretical and practical significance. In order to produce high integration and microminiaturization, all 3D electronic packaging products need to interconnect in the vertical direction [11] the through multi-reflow process, in which the overgrowth of brittle interfacial intermetallic compound (IMC) tends to occur, further resulting in service reliability problems for electronic devices [12–14].

On the one hand, in studies on multiple reflow, most scholars' research into IMC growth in multi-reflow processes simply regards the multi-reflow process as a long-time soldering reaction. This results in the deduction of some inaccurate or mutually contradictory mechanisms for interfacial IMC growth, such as generalized view controlled by

grain boundary diffusion [15–17] or volume diffusion [18,19]. However, the results of our previous studies [20,21] proved that IMC growth kinetics are different at every stage in multiple reflows. In addition, the changes in the mechanical properties of the solder brought about by multiple reflows are also unknown. Further work should be done in order to fully understand the growth of IMC during each reflow and its growth control mechanism, even to the point of being able to control its growth during multiple reflows.

On the other hand, among the studies on interfacial reactions, Cu is the dominant diffusion element at the Sn/Cu interface and the change in Cu concentration directly affects the interfacial reaction, consequently affecting the reliability of solder joints [22,23]. Especially, in the trend of solder joint miniaturization, the volume ratio of IMC to the whole solder joint increases and the effect of Cu concentration changes on the interfacial reaction is more significant [24]. Generally, the microstructure evolution of $Cu_6Sn_5$ grains in the IMC cannot be observed by experimental methods, and most researchers speculate as to its growth process based on experimental results, which can skew their conclusions. In recent years, machine learning and data analytics have been recognized as techniques for the real-time quality monitoring of material processing experiments. For instance, the convolutional neural network (CNN) model was developed to detect $Cu_6Sn_5$ IMC and bubbles in liquid solder [25]; the multi-phase field model was combined with machine learning to assess the growth of $Cu_6Sn_5$ during thermomigration and electromigration process [26,27]; and finite element analysis and machine learning were utilized to predict the morphology of IMC in an Sn-x Ag-y Cu/Cu (SAC/Cu) system [28]. In a word, data-driven science has become a powerful tool to predict the growth behavior of IMC.

This paper explores the variation in Cu concentrations during multiple reflows and its effect on the growth behavior of IMCs and the properties of solder (e.g., hardness and melting point) through a combination of experiments and simulations. The changes in its growth during multiple reflows and the influencing process are described. These results can be used as data combined with correlation results to build machine learning models to predict the evolution of IMC during multiple reflows.

## 2. Materials and Methods

### 2.1. Materials and Experimental Procedure

The diameter and height of the Cu substrate were 1500 µm and 100 µm, respectively. These figures were obtained by using a hole punch. Next, the pure Sn solder ball, with a diameter of 1400 µm, reflowed on the Cu substrate at 250 °C for 30 s for each reflow cycle and was subsequently subjected to air cooling for each cycle. The maximum reflow number was 9 in this study. The schematics of the experimental solder bump and the temperature profile of the nine-time reflow process are presented in Figure 1a,b, respectively. In order to obtain the change in Cu concentration in Sn solder at the isothermal stage, the solder during the reaction was clamped through a homemade clamp.

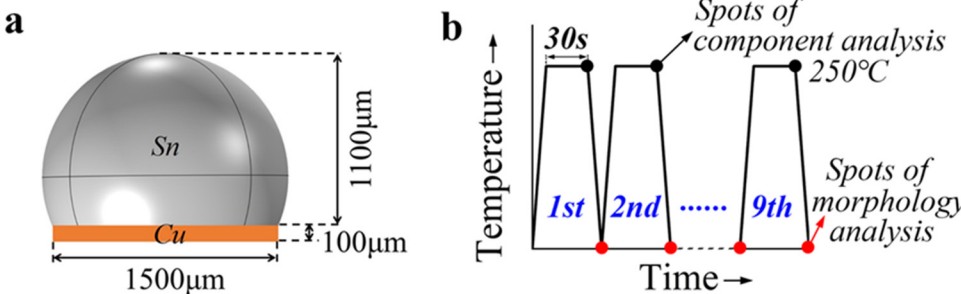

**Figure 1.** Schematics of the experimental solder bump (**a**) and the temperature profile of the nine-time reflow process (**b**).

The top and cross-sectional morphology of $Cu_6Sn_5$ were characterized by scanning electron microscopy (SEM). To expose the top microstructure of $Cu_6Sn_5$, these samples

were etched by 10% $HNO_3$ solution (in volume), while a $5\%HNO_3 + 2\%HCl + 93\%C_2H_5OH$ (in volume) solution was used to obtain the IMC cross-sectional morphology. An electron probe micro analyzer (EPMA) (JXA-8530F PLUS, Japan Electronics, Tokyo, Japan) was employed to detect changes in Cu concentration during the multiple reflow processes. Furthermore, the Vickers hardness and the melting point of the solder were measured by a micro-Vickers hardness tester (HVD-51S, Shanghai Juijing Precision Instrument Manufacturing Co., Shanghai, China) and differential scanning calorimetry (DSC), respectively.

### 2.2. Vickers Hardness Test

With a load of 200~980.7 N, press the square conical diamond with the relative face angle of 136° into the surface of the material, remove the load after holding for a certain time, measure the diagonal length d of the indentation, and then use the formula

$$\mathbf{HV} = 0.102 \times \frac{2F\sin\frac{\alpha}{2}}{d^2} \tag{1}$$

to calculate the surface area of the indentation, and finally the average pressure on the surface area of the indentation, which is the Vickers hardness value of the metal (where alpha is the angle of the indenter relative to the surface, 136°; $\frac{1}{d^2}$ is the reciprocal of the square of the average diagonal length of the indentation, and it is important to note that the unit is $\frac{1}{mm^2}$).

### 2.3. Melting Point Test

In this study, TGA/SDTA851e (Mettler Toledo, Greifensee, Switzerland) equipment was used to obtain the final melting point temperature data from an initial temperature of 50 °C to 300 °C, with a heating rate of 10 °C/min and a cooling rate under argon gas protection.

### 2.4. Numerical Simulation

To clearly understand the change in the Cu concentration in solder during reflow process, a numerical simulation was conducted using COMSOL (Camo Digital Software Technology Co., Stockholm, Sweden) with the finite element method. The solder composite used in the simulation was pure Sn, in which the Cu concentration was initially 0, and the size of solder was 1500 μm × 1100 μm, where 1100 μm is the maximum height and 1500 μm means the wetting diameter on the Cu substrate. In the simulation, three-dimensional geometry and the tetrahedral meshing method were used. For the boundary condition, the reflow temperature was set as 523.15 K and the Cu concentration at the Sn/Cu interface was considered saturated, which is 1.35 wt% for 523.15 K. During the reflow process, the Cu atoms were continuously diffused from the reaction interface into the liquid solder; the diffusion can be calculated by the mass diffusion equation according to Fick's second law,

$$e_a \frac{\partial^2 u}{\partial t^2} + \frac{1}{D} \cdot \frac{\partial u}{\partial t} + \Delta \cdot \Gamma = f \tag{2}$$

and

$$\Delta = \left[ \frac{\partial}{\partial x}, \frac{\partial}{\partial y}, \frac{\partial}{\partial x} \right] \tag{3}$$

where $\frac{1}{D}$ is the reciprocal the average diffusion coefficient of Cu in liquid Sn and the value is set to $1/3.16 \times 10^{-9} s/m^2$ at 523.15; $e_\alpha$ and $u$ represent the mass coefficient and Cu concentration, respectively.

## 3. Results and Discussion

In the interfacial reaction of the reflow of the solder/Cu substrate, two intermetallic compound layers were generated, a $Cu_3Sn$ layer near the Cu side and a scalloped $Cu_6Sn_5$ layer near the solder [29]. This discontinuous scallop-type IMC layer was created by the rapid dissolution of Cu atoms from the substrate into the molten solder. In this paper, the

temperature of each reflow was only 250 °C and the reflow time lasted only 30 s. Compared with the $Cu_6Sn_5$, the thickness of the $Cu_3Sn$ generated at the interface was very small, and the effect was negligible. Therefore, our study is mainly based on the $Cu_6Sn_5$ layer. Figure 2 is the top morphology and cross-section morphology of $Cu_6Sn_5$ during the nine-time reflow process. Based on the top surface morphology of $Cu_6Sn_5$ it is clear that the scallop-like $Cu_6Sn_5$ was distributed uniformly on the whole interface during the first reflow and the second reflow. As the number of reflow cycle increased, not only did the size of the $Cu_6Sn_5$ increase but its morphology also changed, from scallop-like to prismatic. Interestingly, $Cu_6Sn_5$ nanoparticles were found in multiple reflow processes, such as the second reflow and the seventh reflow.

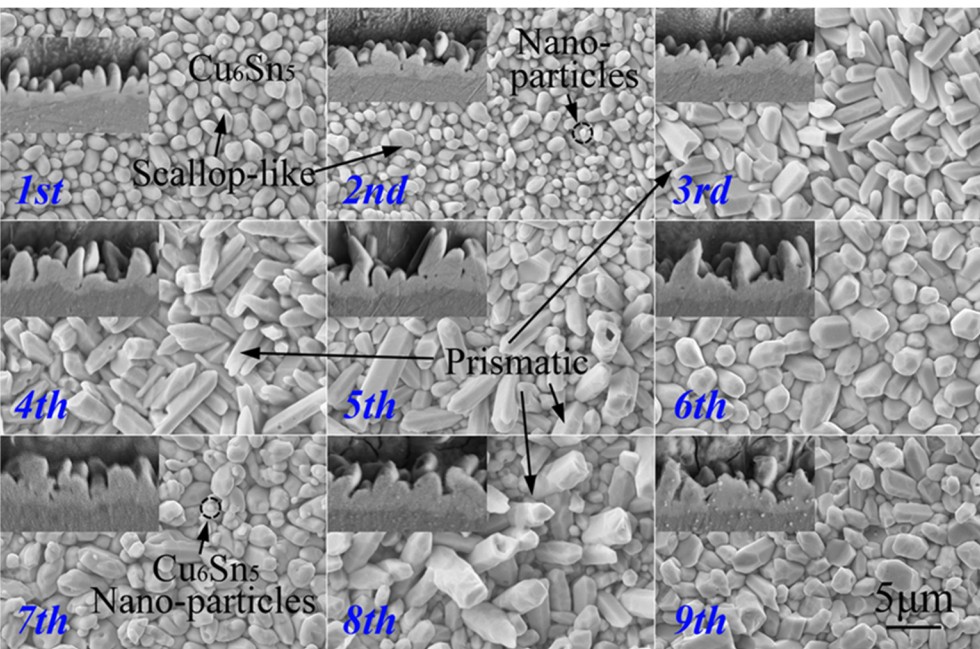

**Figure 2.** Top-view and cross-section morphologies of $Cu_6Sn_5$ during the nine-time reflow process.

Since the diffusion rate of Cu is faster than Sn, the growth behavior of $Cu_6Sn_5$ is mainly controlled by Cu diffusion [22,23,30]. Figure 3 is the Cu mass fraction in solder obtained by experiment (a) and simulation (b) (for further details, see part I in Supplementary Material). For each sample, three points were selected for measurement to obtain the average value of the Cu mass fraction. Figure 3c presents the plots of the measured and calculated Cu mass fractions. Clearly, the Cu mass fraction increased as the number of reflow cycles increased. In the process of multiple reflow, Cu atoms continuously diffused from the substrate into the liquid solder, resulting in the increase in the Cu concentration in the solder joint.

In recent studies [31,32], the growth mechanism of $Cu_6Sn_5$ was different in the isothermal stage and cooling stage and the formation of the final morphology and the growth of $Cu_6Sn_5$ mainly occurred at the cooling stage. The increases in the Cu concentration in liquid Sn solder at the isothermal stage led to the increased deposition of Cu (in the form of $Cu_6Sn_5$ clusters) at the cooling stage, resulting in increases in the $Cu_6Sn_5$ grain size. Furthermore, $Cu_6Sn_5$ is essentially a hexagonal prism growth form, which is formed by the deposition of $Cu_6Sn_5$ clusters at scallop-like $Cu_6Sn_5$ [31,33]. For the first and the second reflow cycle, the Cu concentration in the liquid Sn solder is not enough for the growth of prismatic. This is why the morphology of $Cu_6Sn_5$ changed from scallop-like to prismatic as the number of reflow cycles increased. It is worth noting that there was some deviation between the measured data and the simulated data. As demonstrated in Figure 3b, the concentration of Cu in solder fluctuated, increasing with the increase of reflow times. For the simulation, the result was the behavior of Cu diffusion under ideal conditions. However, the growth of $Cu_6Sn_5$ was a process of continuous dissolution and

growth in the actual reaction process. The dissolution and growth of $Cu_6Sn_5$ during the reaction process resulted in the fluctuation of Cu concentration in the reaction process. A sketch of the growth behavior of $Cu_6Sn_5$ during multiple reflows is presented in Figure 4. The results provide a data basis for the data-driven prediction of $Cu_6Sn_5$ growth during multiple reflow.

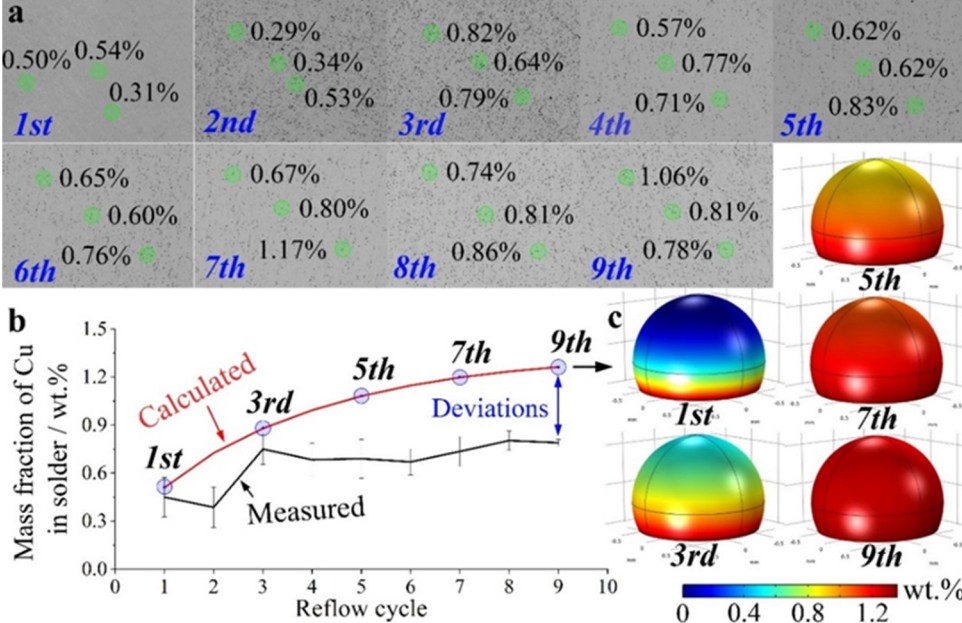

**Figure 3.** EPMA results for the Cu mass fraction in solder (**a**), plots of the measured and calculated Cu mass fractions (**b**) and simulations for the Cu distribution in solder balls (**c**) during the nine-time reflow process.

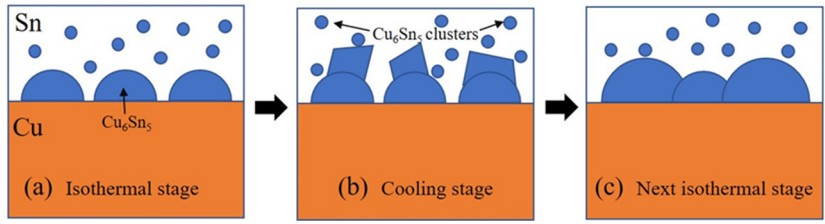

**Figure 4.** Sketch for the growth behavior of $Cu_6Sn_5$ during multiple reflows. (**a**) Isothermal stage, (**b**) Cooling stage, (**c**) Next isothermal stage.

Figure 5a is the evolution of IMC thickness and growth rate. For the first reflow, the growth of IMC achieved the fastest growth rate. Furthermore, the IMC layer formed at the first reflow inhibited the diffusion of Cu atoms in the subsequent reflow process, resulting in a decrease in the growth rate. Since the growth of the IMC was closely related to the deposition of $Cu_6Sn_5$ clusters at the cooling stage and the IMC layer dissolved in the next reflow process, the fluctuation of the Cu concentration in the solder led to a change in the growth rate, which further led to fluctuations in the IMC thickness.

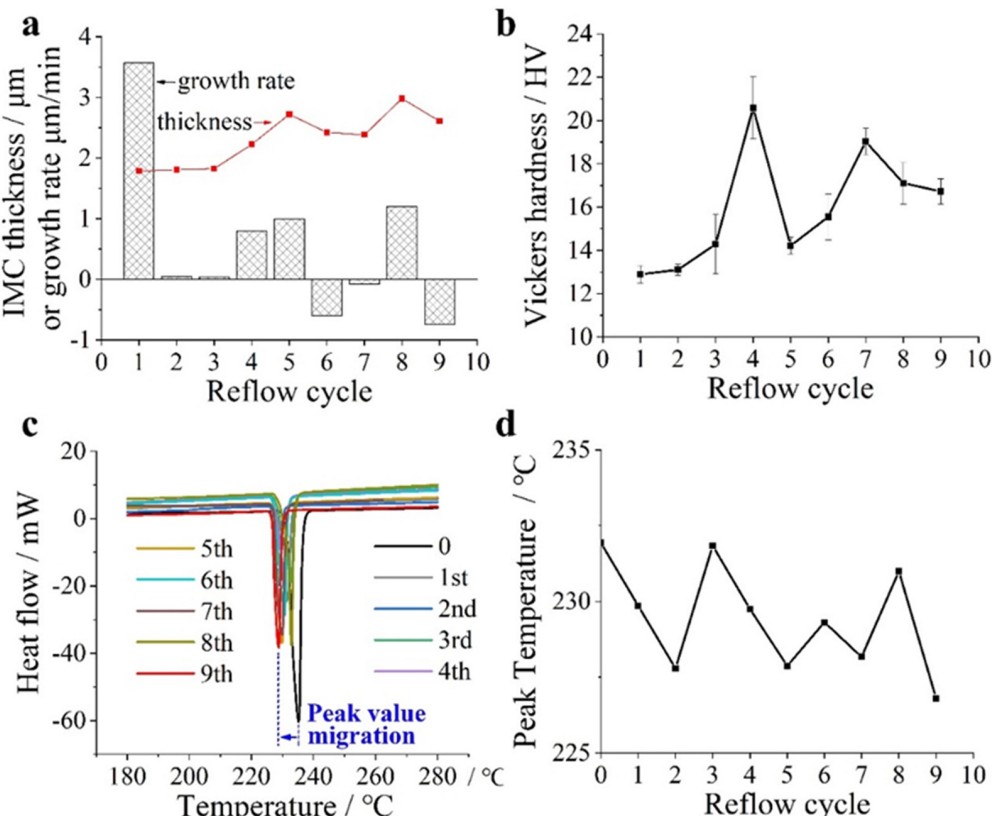

**Figure 5.** Evolutions of IMC thickness and growth rate (**a**), Vickers hardness of the solder bumps (**b**), DSC results (**c**) and melting points of the reflowed solders (**d**) during the nine-time reflow process.

Furthermore, the fluctuation in the Cu concentration in solder during multiple reflow affected the solder properties, such as Vickers hardness and the melting point (for further details, see part II in Supplementary Material). For the Vickers hardness, five different points were selected to measure for each sample, as presented in Figure 5b. Clearly, the hardness of the solder fluctuated during multiple reflows and the highest value was obtained at the fourth reflow. Combined with the Cu concentration presented in Figure 3, the Cu concentration was close to its eutectic point at the fourth reflow, where it displayed its highest hardness. Figure 5c,d feature the DSC results and melting points of the solder during the nine-time reflow process, respectively. Clearly, the melting point of the solder fluctuated during the multiple reflow processes due to the fluctuation in the Cu concentration. Overall, a slow downward fluctuating trend was observed.

## 4. Conclusions

In summary, an experimental measurement was combined with a simulation calculation to investigate the evolution of the Cu concentration in solders during multiple reflows, as well as its effects on the growth behavior of IMC and solder properties. The key points of this research can be summarized as:

1. The concentration of Cu in solder fluctuates, increasing with the increase in reflow times, which is closely related to the continuous growth and dissolution of IMC.
2. Not only does the size of $Cu_6Sn_5$ grains increase, but its morphology also changed from scallop-like to prismatic as the number of reflow cycles increased. The changes in the Cu concentration in solder during the multiple reflows resulted in fluctuation in the growth rate of the $Cu_6Sn_5$.
3. The Vickers hardness and melting point of the solder fluctuated during the multiple reflow processes due to the fluctuation in the Cu concentration.

**Supplementary Materials:** The following are available online at https://www.mdpi.com/article/10.3390/met11122041/s1, Table S1: Data of the calculated and measured Cu mass (wt%) fraction in solder; Table S2: State of the IMC growth rate, IMC thickness, Solder melting point and HV value in each reflow.

**Author Contributions:** M.S.: data curation, formal analysis, methodology, writing—original draft. C.D.: data curation, software, methodology. H.M. (Haoran Ma): writing—review and editing. Y.W.: writing—review and editing. H.M. (Haitao Ma): funding acquisition, writing—review and editing. All authors have read and agreed to the published version of the manuscript.

**Funding:** This research was funded by the National Natural Science Foundation of China (Grant No. 51871040) and the National Science Foundation for Young Scientists of China (Grant No. 52101035).

**Institutional Review Board Statement:** Not applicable.

**Informed Consent Statement:** Not applicable.

**Data Availability Statement:** The raw/processed data required to reproduce these findings cannot be shared at this time due to technical or time limitations.

**Conflicts of Interest:** This manuscript has not been published elsewhere and is not under consideration for publication in any other journal. All the list authors have approved the manuscript and its submission to the journal.

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
