# Peer review of "Simulation for Cu Atom Diffusion Leading to Fluctuations in Solder Properties and Cu6Sn5 Growth during Multiple Reflows"

_metals, doi:10.3390/met11122041_

Round 1

Reviewer 1 Report

The authors performed study of the change of Cu concentration during multiple reflow and its impact on the growth behavior of IMC and the property of solder, such as hardness and melting point.

The results are important for the improvement of soldering technologies, thus they are new and not only of scientific interest but also of practical application. The manuscript is concise enough, clearly presented and well organized, the English is satisfactory.

At the same time, there are several points that require clarification or explanation, namely:

  • The authors analyze only the intermetallic Cu6Sn5 layer, while the interaction between Sn and Cu results in sequential formation of two layers Cu3Sn and Cu6Sn5 (you can refer to e.g. https://doi.org/10.1007/s10854-017-6877-7  and explain).
  • Each subsequent reflow leads to an increase in the IMC layers and, consequently, to the content of copper, but since pure copper is not detected, it is probably more correct to talk about the growth of the IMC layers, rather than the amount of copper.
  • Fig 3a presents three points, selected to obtain the average value of Cu mass fraction, but only one value at the 4th reflow is close to the eutectic composition (0,7 wt. % Cu). Therefore, it can hardly be unambiguously related to hardness, which is almost the same as after reflow No7.    
  • A change in the melting temperatures is important for soldering processes, so, it would be useful to give the numeric Tm values additionally to plots in Fig. 4c.
  • Since the results and the discussion are combined into one section 3, the section “4 Discussion” seems redundant.

The paper could be accepted after some revision.

Reviewer 2 Report

  1. The title should be defined more accurately. The authors performed a specific simulation of the fluctuation process. They did a specific simulation study. The current title may show that the review has been performed, but not the results of a specific analysis.
  2. The keywords are wrongly chosen. We start with the most important to the less important. According to the keywords, searches are performed in the databases.
  3. Introduction - that is, the literature review is poorly done.
  4. Why is Figure 1 in Chapter 3. The structure of the publication is bad.
  5. The modeling in the program is too poorly described. If no experimental tests were performed, on what basis was the model validated? Chart 2 shows the values from the measurement - it is not clear whether from the program or the experiment?
  6. How were the HV hardness values obtained?
  7. The publication is badly prepared. Must include test and simulation details. 

Round 2

Reviewer 1 Report

The paper has been improved and can be published. The only question still remains, namely, about sequential formation of two layers Cu3Sn and Cu6Sn5 (see below). 

"...The authors analyze only the intermetallic Cu6Sn5 layer, while the interaction between Sn and Cu
results in sequential formation of two layers Cu3Sn and Cu6Sn5 (you can refer to e.g.
https://doi.org/10.1007/s10854-017-6877-7 and explain)."

Reviewer 2 Report

The text has been improved. However, I believe there is still room for improvement. Nevertheless, it may be allowed to be published. The reviewers' comments are intended to ensure the appropriate quality and scientific quality of the publication. 
